# Longitudinal Study to Assess the Quantitative Use of Fundus Autofluorescence for Monitoring Disease Progression in Choroideremia

**DOI:** 10.3390/jcm10020232

**Published:** 2021-01-11

**Authors:** Adam M Dubis, Wei S Lim, Jasleen K Jolly, Maria Toms, Robert E MacLaren, Andrew R Webster, Mariya Moosajee

**Affiliations:** 1NIHR Biomedical Resource Centre at Moorfields Eye Hospital and UCL Institute of Ophthalmology, London EC1V 9EL, UK; a.dubis@ucl.ac.uk (A.M.D.); andrew.webster@ucl.ac.uk (A.R.W.); 2Department of Genetics, Moorfields Eye Hospital NHS Foundation Trust, London EC1V 2PD, UK; weisinglim1@gmail.com; 3Nuffield Laboratory of Ophthalmology, Department of Clinical Neurosciences, University of Oxford, Oxford OX3 9DU, UK; enquiries1@eye.ox.ac.uk; 4Oxford Eye Hospital, Oxford University Hospitals NHS Foundation Trust, Oxford OX3 9DU, UK; enquiries@eye.ox.ac.uk; 5Development, Ageing and Disease, UCL Institute of Ophthalmology, London EC1V 9EL, UK; m.toms@ucl.ac.uk; 6Great Ormond Street Hospital for Children NHS Foundation Trust, London WC1N 3JH, UK

**Keywords:** inherited retinal disease, autofluorescence, retinal imaging, longitudinal clinical study

## Abstract

Background: Characterisation of preserved autofluorescence (PAF) area in choroideremia (CHM) and its validity for monitoring disease progression in clinical trials is of importance. Methods: Eighty patients with molecularly confirmed CHM were recruited. PAF area was measured manually by 2 graders and half-life was calculated based on exponential decay model. Results: Mean age at baseline and follow-up examination was 38.1 (range, 10–69) and 40.7 (range, 11–70) years. Mean follow-up interval was 29 months (range, 6–104). The median LogMAR visual acuity was 0.10 (OD) and 0.18 (OS). Interobserver repeatability for PAF area was −0.99 to 1.03 mm^2^ (−6.46 to 6.49% of area). There was a statistically significant relationship between age and rate of PAF area loss (r^2^ = 0.28, *p* = 0.012). The half-life for PAF area was 13.7 years (range, 1.7–216.0 years). The correlation between half-life and age was stronger than between half-life and log transformed baseline PAF area, although neither was statistically significant. Conclusions: The intra- and inter-observer PAF area measurement variability provides a baseline change, which must be overcome in a clinical trial if this metric were to be used. Treatments must slow progression to alter the exponential decay in a timely manner accounting for naturally slow progression patterns.

## 1. Introduction

Choroideremia (CHM) is an X-linked chorioretinal dystrophy, characterised by progressive atrophy of the retinal pigment epithelium (RPE), outer retina, and choroid. It results from mutations affecting the *CHM* gene, which is located at Xq21.2 comprising 15 exons and encodes an ubiquitously expressed protein containing 653 amino acids; Rab escort protein 1 (*REP1*) [1]. REP1 acts as a facilitator of post-translational lipid modification (prenylation) of Rab proteins permitting intracellular vesicular transport [2].

The majority of mutations in the *CHM* gene result in a complete loss of REP1 protein function [3]. Over one-third of patients harbour nonsense mutations, 25–50% represent full gene and partial deletions [4]. The mutation spectrum extends to cover insertions, duplications, translocations, splice-site, frameshift, missense, promotor, and deep intronic mutations [5,6,7,8,9]. The corresponding CHM phenotype has exhibited high inter- and intra-familial variability [10]. To date there has not been a strong retinal phenotype/genotype correlation; however, some studies have begun to link transcript level to phenotype [11,12]. Affected male patients typically develop night blindness during childhood, followed by restriction of the peripheral visual field during early adulthood, and finally a decrease in central visual acuity, often leading to complete blindness by the 5–6th decade of life [13]. The emergence of human gene therapy clinical trials calls for suitable imaging techniques for monitoring disease progression in order to assess treatment effects.

Fundus autofluorescence (FAF) is largely a byproduct of lipofuscin deposits in the RPE, which are derived from incomplete degradation of phagocytosed photoreceptor outer segments. Lipofuscin is not inherently autofluorescent, but rather contains various biomolecules including lipids, protein, and retinoids (A2E/A2E*) [14,15]. The pattern of FAF varies in males with choroideremia depending on the stage of the disease. In younger patients, the FAF pattern can be generalised as an intact central island of preserved autofluorescence (PAF) area, with mottled areas of reduced autofluorescence peripherally [10,16]. In advanced stages of the disease, areas of chorioretinal atrophy correlate with dramatically reduced PAF area, with small residual islands of FAF separated from the main central island [16,17,18,19]. Several groups have already employed cross-sectional FAF analysis for CHM both for natural history and gene augmentation studies. Cideciyan et al. looked at FAF patterns in CHM based on different types of FAF imaging (standard short wavelength, reduced power AF and near infra-red AF) [20]. This study illustrated that FAF was versatile, as similar patterns were observed under all imaging conditions. Jolly et al. [18] characterised the patterns observed in a consecutive set of clinical patients, observing patterns of FAF constriction. Their work suggested a logarithmic decline over age, but as the cohort was cross-sectional this could not be tested. MacLaren et al. [21] and Xue et al. [19] both presented cross-sectional analysis of FAF areas in relation to a cohort undergoing gene replacement therapy for CHM. Xue and colleagues showed a strong correlation between PAF area and both outer nuclear layer (ONL) and choroidal thicknesses, and Jolly et al. [18] demonstrated strong correlations between FAF area and microperimetry sensitivity, further suggesting FAF as a significant and repeatable measure for a clinical trial. More recently, Aylward et al. [5] performed a two year longitudinal study in 31 younger patients and found that the rate of degeneration in terms of percentage FAF loss per year was constant at all ages, following an exponential decay. More recent work by Hariri et al. [22] established a common measurement area and nomenclature in a multicentre cohort of CHM patients. Another pair of recent meta-analysis studies by Shen et al. shows that while visual acuity reduces in a bilinear pattern [23], PAF area is more complex with residual area following a log transformed decline (exponential) [24]. Both studies follow patients over long time periods not conducive to modern trial time scales. While these studies make strong suggestions at the robustness of PAF area measurements, the follow-up time interval must also include data on intra- or inter-observer variability in order to provide the most accurate assessment of disease progression.

This report describes longitudinal visual acuity and PAF area changes associated with the genetic mutations found in 80 patients from 76 families with CHM in the UK, in order to suggest a viable method of monitoring the outcome of prospective gene therapy trials for choroideremia.

## 2. Experimental Section

### 2.1. Subjects

A cohort of 80 patients from 76 families with a clinical diagnosis of CHM were obtained from Moorfields Eye Hospital (ARW and MM) and Oxford Eye Hospital (REM). After informed consent was obtained, blood samples were taken for DNA extraction and mutation screening of the *CHM* gene. Sequencing for coding and non-coding mutations followed standard protocols [25]. 

### 2.2. Clinical Evaluation

Clinical evaluation included best-corrected visual acuity (BCVA), which corresponds to the most recent follow-up visit, in logarithm of the minimum angle of resolution (LogMAR), any Snellen acuity measurements were converted to the equivalent LogMAR; colour fundus photography and fundus autofluorescence imaging.

### 2.3. Fundus Autofluorescence (FAF) Imaging

The FAF imaging was performed using a confocal scanning laser ophthalmoscope (cSLO). From 2003 to 2009, images were obtained using an HRA2 (excitation light 488 nm, barrier filter 500 nm, field of view 30° × 30° Heidelberg Engineering GmbH, Heidelberg, Germany) [26]. After 2009, images were obtained using the Spectralis with viewing module version 5.1.2.0 (excitation light 488 nm, barrier filter 500 nm, field of view 30° × 30°, or 55° × 55° Heidelberg Engineering GmbH) [27]. Patients had all sessions imaged using the same field of view to allow for comparison. The standard corneal K values provided in the software (7.7 mm) were used in all cases.

Inclusion criteria for this study were that gradable FAF images were obtained on at least two occasions separated by at least nine months, and they needed to be of the same field of view. Gradeability was determined as the consensus between two graders. All eyes were analysed but only the eye with largest baseline PAF area was used for subsequent analysis due to the high degree of interocular symmetry in these patients. The interval of observation was determined by the difference between the baseline and most recent follow-up FAF imaging in years (y).

The PAF area was defined as the area of the central island only, and did not include any detached PAF islands area. The PAF area was measured using the area measurement tool available on the Heidelberg software, which reported in mm^2^. All measurements were collected by two investigators (WL, JKJ, adjudicated by AMD). The examiner manually traced the boundary of intact AF and the software calculated the size of the area marked (Figure 1). The average PAF area half-life (years) was calculated based on the fitted exponential function for all individual subject data points within the dataset. Comparisons of PAF area were compared on log-transformed baseline PAF area (log-mm^2^).

In order to determine the variability in measurements between different observers, JKJ and WL both measured the intact FAF area for 68 eyes from 36 patients. Poor image quality images causing a mismatch between eyes and patients were removed. In order to determine the variability of measurements by the same observer, WL repeated measurements of 51 FAF images from 30 patients taken from the intra-grader image set. As with inter-observer measurements, some images were removed due to poor image quality. All images for repeatability came from the larger cross-sectional study.

### 2.4. Statistical Analysis

Bland–Altman analysis was used to analyse the intra-observer (WL only) and inter-observer (WL vs. JKJ) variability of the measurements. Areas were skewed, and a log transformation effected normality. PAF area half-life was plotted as a function of age and log transformed baseline PAF area measurements. These relationships were assessed for statistical significance using JMP13 (SAS Institute, Cary, NC, USA). Linear regressions were compared using nonparametric tests (F-statistics) for significance and comparison between groups was done using the nonparametric Fischer’s Exact test. Statistical significance was set at *p* < 0.05.

## 3. Results

### 3.1. Clinical Findings

Our study cohort included 80 patients with a clinical and genetic diagnosis of CHM. The clinical and molecular findings are summarised in Appendix A. All patients were male, 75 were Caucasian (94%) and 5 were South Asian (6%). The mean age at baseline and follow-up was 38.2 and 40.7 years (range, 10–69 and 11–70), respectively. The mean follow-up interval was 29.0 months (range, 6–104). Five patients (6%) presented before 16 years of age, mean 13 years (range, 10–15), and 75 patients (94%) presented age 16 years or over. The most recent median LogMAR visual acuities were 0.45 (range, −0.08–2.70) in the right eye and 0.35 (range, −0.08–2.70) in the left eye. The majority (OD = 63%, OS = 55%) of patients maintained good visual acuity levels <0.20 in both eyes until late in the disease (Figure 2a).

### 3.2. FAF Findings

We were interested to know how the retinal area loss correlated between eyes (the degree of interocular symmetry) as well as seeing how the rate varied across the cohort. There were significant (*p* < 0.0001) correlations between eyes at baseline (r^2^ = 0.863) and at follow up (r^2^ = 0.861). Only one eye from one patient did not have usable data at both time points. For the right eyes, the mean size of intact central PAF area at baseline and follow-up was 15.2 mm^2^ (range, 0.19–76.96) and 12.1 mm^2^ (range, 0.1–58.9), respectively. For the left eyes, the mean size of the intact central PAF area at baseline and follow-up was 16.13 mm^2^ (range, 0.1–85.4) and 12.6 mm^2^ (range, 0.05–67.8). The rate of PAF area decrease ± SEM for right eyes was 1.8 ± 0.48 mm^2^/y (range: 22.92 to 0.01 mm^2^/y) compared to 2.15 ± 0.54 mm^2^/y (range: 55.6 to 0.01 mm^2^/y) for the left eyes. Overall, the rates of loss between eyes were highly correlated (r^2^ = 0.84, Figure 2b). Due to the high degree of correlation between eyes, when only the eye with highest baseline PAF area was analysed longitudinally, resulting in 80 eyes being used for assessing trends over time. There is a high correlation between baseline and follow up PAF area (r^2^ = 0.96; Figure 2c). PAF area by age is shown in Figure 2d. There was an exponential decline in PAF area with age (r^2^ = 0.20, *p* = 0.0012). The exponential decline was described by the function p(t) = p(0) × e^(kt)^, where p(0) was 38.829, and k was −0.041. This function provided a half-life of residual PAF area of 13.7 years. However, the strongest predictor of rate of PAF area decrease was size of PAF area at baseline (r^2^ = 0.56, *p* < 0.0001).

Interestingly, in this dataset seven patients had remarkably slow degenerations, with half-lives over 30 years. This slow period of degeneration could not be described by age, baseline PAF or mutation type. Figure 3 shows the impact these seven subjects have on the cohort trend when comparing half-life to log-transformed baseline PAF area (a) and age (c). As vision loss reaches final stages in the 4 or 5 decades of life, data was filtered for half-lives over 30 years. This filtered data is shown in Figure 3 (b–log transformed baseline PAF area, d–age). Filtering improved the relationship for age (r = 0.12 vs. 0.13) but decreased the fit for log transformed baseline PAF (r = 0.11 vs. 0.06). None of these relationships rose to the level of statistical significance.

### 3.3. Intra- and Inter-Observer Measurement Repeatability

The Bland–Altman plot was used to determine the variability of the measurements carried out by a single grader (WL). Fifty-one eyes in total (a mix of right and left) were graded twice, with 95% of the differences between readings ranging from −0.99 to 1.03 mm^2^. The bias between gradings was 0.017 mm^2^, suggesting the grader was highly repeatable. More importantly, these numbers reflect all sizes of intact PAF area. If the difference is expressed as percent of central PAF island area the 95% confidence interval is −6.4% to 6.5% with negligible bias (0.014). In order to investigate inter-observer variability, the Bland–Altman plot was used to calculate the disagreement of measurements carried out by two different graders (JKJ and WL, adjudicated by AMD). Sixty-eight images were graded by each observer. There was a slight bias between observers (−0.28 mm^2^) with a 95% confidence interval of −1.74 to 1.18 mm^2^. When this was transformed to percent of full central PAF island area, the data showed a customary deviation at small areas (small error results in a large percent difference on a small area compared to large area). The bias between observers is −0.87%, confidence interval is −17.3 to 15.6%. The distributions of all repeatability metrics can be seen in Figure 4.

### 3.4. Genotype-FAF Phenotype Correlation

Molecular genetic testing of the coding region, intron-exon boundaries, and western blot analysis confirmed mutations involving the *CHM* gene in all 80 patients with 47 deletions (involving 6 whole gene, 17 exonic, 24 several base pairs), 22 in-frame nonsense, 2 insertions, 3 duplications, 11 splice site, and 1 missense mutations (Appendix A). Seven were novel mutations, including deletions, splice site, and a nonsense mutation, highlighted in Appendix A. To determine a genotype–phenotype correlation, data from the greatest PAF area at baseline was compared for each mutation type. There was no difference in the slope of PAF size by age (*p* = 0.403) or rate of PAF area of atrophy by age (*p* = 0.529) between the three mutation types when stratified by deletion, missense/nonsense and splice site changes (Figure 5).

## 4. Discussion

Choroideremia is a progressive degeneration of the choroid, RPE, and photoreceptors that typically spares the macula until late stages, thus, ensuring good central visual acuity over many years [23,28]. The CHM phenotype presents with high variability. Most studies so far have been cross-sectional, thereby limited in their investigation into the progressive nature of disease [29,30]. This study assessed longitudinal changes in RPE atrophy by undertaking FAF imaging in a large well-characterised cohort of patients with CHM and assessed the repeatability of these measurements. Shen et al. highlighted the need for longitudinal studies to provide key parameters for use of FAF in clinical trials. The findings herein assist in providing improved advice on prognosis and may inform patient follow-up intervals for future therapeutic interventions.

The majority of mutations identified in this study are null, resulting in the absence of the gene product REP1 owing to complete loss of the gene or nonsense/frameshift deletions. We found one family with a known missense mutation H507R, which produces an inactive REP1 variant that cannot interact with Rab geranyl-geranyl transferase and, therefore, excludes it from the isoprenylation cycle [1,6]. Of the 80 mutations, seven were novel to our cohort compared to all those listed in HGMD [8,9]. All the mutations were evenly distributed across the whole gene. There was no overall genotype-phenotype correlation seen between any particular mutation type, hence, this cannot be used a prognostic indicator of disease progression.

Despite accumulating knowledge of the genetics of CHM, the actual pathogenetic mechanism of the combined retinal, RPE, and choroidal degeneration remains ill-defined, but is thought to be due to a deficiency in the prenylation of multiple Rab proteins [3,13]. Advanced retinal imaging modalities can provide new findings on the pathophysiology of CHM. Jacobsen et al. [31] studied 21 CHM male patients, aged 6–61, with confocal scanning laser ophthalmoscopy. Two studies have looked at multicentre datasets [22,24], which provided insights into repeatability [22] and longitudinal population metrics [24]. The exponential decline model was put forward by Aylward et al. [5], however, they also found a log-linear relationship when comparing area only. Shen et al. [24] used mixed methods containing both longitudinal and modelled longitudinal data. While their model for creating longitudinal data contained several parameters, as applied in Figure 2d and Figure 5 in this manuscript, the parameters still do not accurately predict the variability present in real life degeneration shown in this manuscript. It is possible that the datasets defer, but since several of the patients used in this study are likely present in their study, this is unlikely. A second possibility is that when modelling data some metric for interpolation is required, which usually includes a term describing the mean change, thus, forcing data away from natural population variation and towards a mathematical mean and resulting in a skew to any global trends observed in un-modelled data. Importantly, past degeneration phenotype still seems to be the strongest predictor of future progression, i.e., patients who present with mild disease regardless of age of presentation, remain on a mild progression trajectory. If this is removed from the analysis through the use of a correction factor, this key relationship is lost. The recent larger cohorts of data, which have been aggregated via meta-analysis [24] or data sharing [22], have provided great insight into the average population dynamics of choroideremia progression. However, there are seven patients shown in Figure 3 that do not follow the trend. Baseline measurement, age, and mutation did not predict this behaviour, and this slow decline and elongation visual function beyond the 5^th^ decade of life have been observed [22]. This individual variance will need to be accounted for, potentially by a one-year observational lead into a therapeutic trial rather than starting from baseline, supported by Figure 2c, suggesting that a person’s previous decline most closely matches their future natural history regardless of follow up interval.

It is important to discern the accuracy of PAF area measurements not only between observers but compared to the same observer. We found significant intra- and inter-observer variability, particularly in the late stages when PAF areas were very small, which would impact on the interpretation of the change of PAF area detected. Interestingly, our data showed a characteristic trend of size dependent repeatability variance. This was not observed in the work by Hariri et al. [22]. This difference highlights the need for highly skilled graders and photographers if performing this task manually as it may be due to variable image quality. The boundary of atrophic lesions on FAF images can be challenging to discern in the greyscale, and the overall quality of the images at different time points will also affect the measurements taken. Automated software, which defines and calculates the atrophic lesions may arguably be more reliable in performing quantitative analysis, although this also is not without its limitations. In addition, ultra-widefield can effectively be helpful to measure the area of atrophy, despite some corrections being needed to correct distortions of the peripheral field. Despite the observer error, clinical variability exists as demonstrated by the outliers, and further investigations are required, especially between those from the same family, to determine the modifying factors.

A further potential limitation of this study, inherent to retrospective studies, is the variable number and interval of examinations during follow-up. Prospective studies with steady additional time point sampling, for example six monthly, would yield more consistent data but may not be needed given the general slow decline of PAF area in this cohort. To maximise effect, selection of younger patients undergoing greater area decline per year would have the greatest chance of showing significantly slow decline, above the margin provided by repeatability. This conclusion is further supported by the strong relationship of repeatability as a function of percent size, with error being smaller for larger areas. This is largely due to the fact that individual pixels correspond to less of the entire area in larger PAF areas compared to smaller residual areas. The significant phenotypic heterogeneity in this disease population adds to the challenges in establishing treatment and control cohorts [32]. This is highlighted by the slower decline observed in this cohort compared to some recent publications [5,16], which could be due to a cluster of patients close to 20 years of age influencing the trend (Figure 2d). This study supports the use of PAF area as a metric for monocular treatment due to the high correlation between eyes for all parameters, also providing evidence that the fellow eye could be used as a natural history marker by which to ascertain change/progression.

## 5. Conclusions

This longitudinal study has investigated the use of PAF area to determine the progressive changes of RPE atrophy in choroideremia. The data suggest there is an association between PAF area and age. The rate of PAF area decrease for younger patients may be higher, however, these images were excluded, as the boundaries of the PAF area were outside the imaging window, hence, this will affect the results seen in this age group. This information will assist genetic counselling and clinicians providing natural history information to patients. PAF area proves to be useful in monitoring the rate of decline over long periods of time (> 12 months minimum), but due to the high levels of inter- and intra-observer measurement variability its use over the short term for assaying treatment responses may not be suitable. A multimodal approach would be recommended for deep phenotyping patients including the use of microperimetry with FAF overlay using programs such as the Alignment GNU Image Manipulation Program version 2.8.10 used in a recent study to correlate the retinal structure and function of female CHM carriers [33]. As gene therapy trials for choroideremia have been initiated, the interpretation of FAF changes must be considered as a useful indicator of outcome, albeit over the long-term.

## Figures and Tables

**Figure 1 jcm-10-00232-f001:**
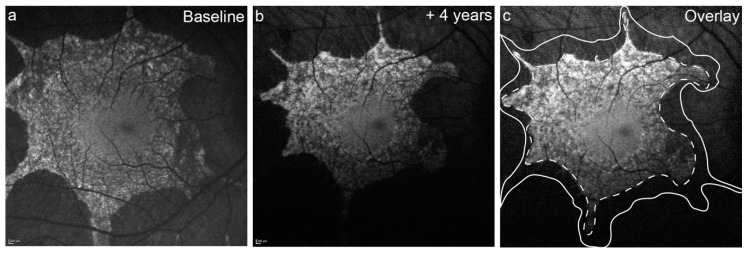
Rate of preserved autofluorescence (PAF) area loss seen in fundus autofluorescence (FAF) images. Measurement of the PAF area at baseline (**a**). The foveal area remains intact with consistent PAF, surrounded by an area of mottled AF. The second follow-up timepoint is from four years later (**b**). The baseline PAF area is shown as the solid line, while follow up PAF is a dashed line (**c**). Alignment of the patient to camera while imaging can cause differences in illumination across images (**a** to **b**), which may lead to differences in measurement sizes.

**Figure 2 jcm-10-00232-f002:**
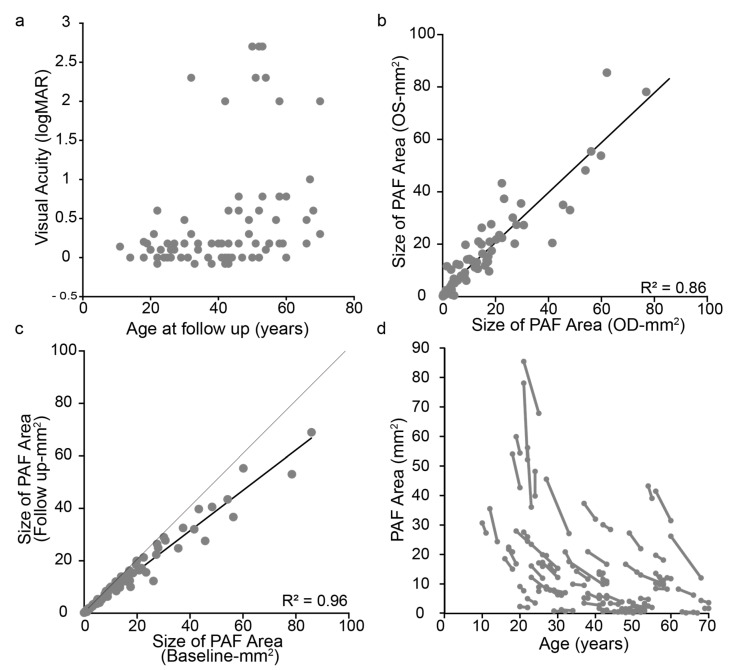
Graphical representation of best corrected visual acuity (BCVA), PAF area (mm^2^) against age (y). (**a**) shows the relationship between BCVA (LogMAR) and age (years) in right and left eyes; (**b**) shows the agreement between eyes at baseline, which is also indicative of the agreement at follow up (data in Appendix A). Hence, largest eye at baseline was selected for further analysis; (**c**) illustrates the agreement in PAF size at baseline compared to follow up, with the light grey line signifying the trend if no change was observed and the dark line is representing the data trend; (**d**) shows the change in PAF area for all subjects as a function of age. Each subject is represented by a dot with connecting line, the relationship showing the exponential decline.

**Figure 3 jcm-10-00232-f003:**
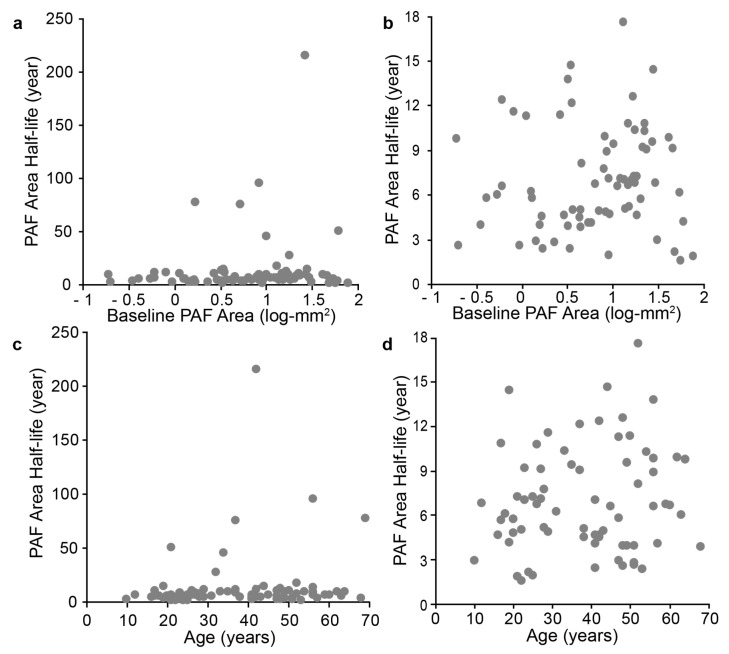
Relationship between PAF area half-life and progression. Panels (**a**) and (**c**) show the raw relationship between PAF area half-life and log transformed baseline PAF area ((**a**), r = 0.11) and age ((**c**), r = 0.13). Due to several subjects showing remarkably slow progression over the observation period, their half-life values were greater than 50 years, as this is not physiologically viable over the course of the disease, the data was filtered to remove half-life greater than 30 years. These filtered results are shown in panels (**b**) (log transformed baseline PAF area) and (**d**) (age). The data fit reduced to r = 0.07 for log transformed baseline PAF area but improved for age (r = 0.14).

**Figure 4 jcm-10-00232-f004:**
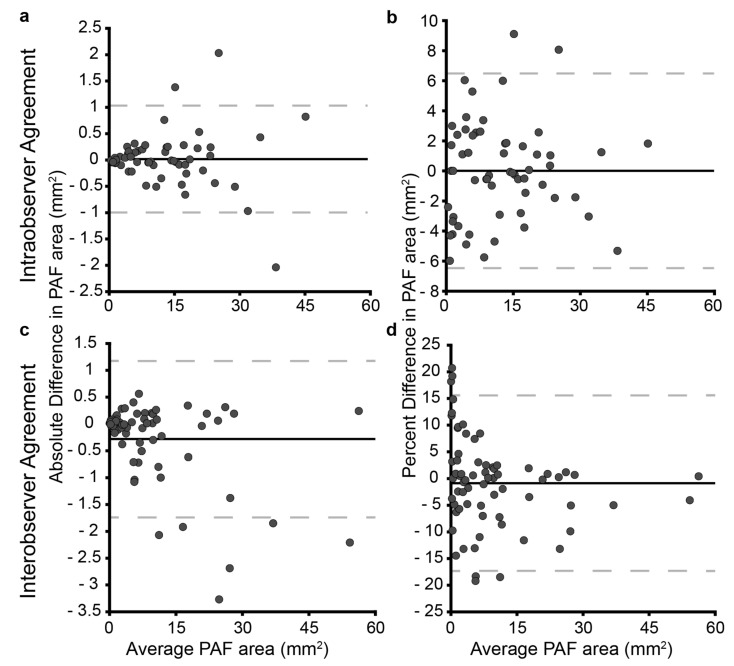
Agreement within and between graders for manual measurement of PAF area. (**a**,**c**) show the absolute difference between gradings ((**a**), intraobserver and (**c**), interobserver) while (**b**) (intra-) and (**d**) (inter-) show percent of full PAF area difference. As the PAF area gets smaller, a small difference in measured area (mm^2^) equates to a larger difference in percent of residual PAF, therefore, in a clinical trial change in terms mm^2^ needs to be calibrated by the size of the circumference to determine accuracy.

**Figure 5 jcm-10-00232-f005:**
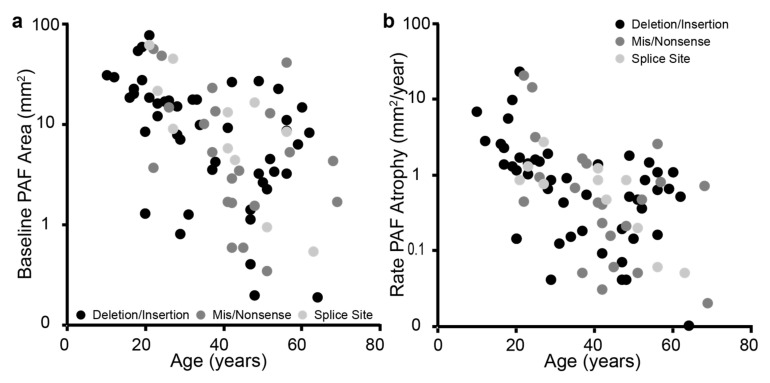
Assessing the relationship between mutation type and disease severity. (**a**) Shows the relationship between size of the PAF area at baseline with age for deletion/insertion mutations (black), mis/nonsense mutations (dark grey), and splice site mutations (light grey). (**b**) Displays the relationship between the rate of PAF area decrease to age for the three mutation types. There were no statistically different slopes with regards to size of PAF area or rate of decrease.

## Data Availability

All measurements generated for this publication are available in the Appendix A. Due to the identifiable nature of retinal images in rare diseases, images cannot be made freely available, but can be made available upon reasonable request and obtaining the appropriate legal and ethical approvals.

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
