# Peer review of "Longitudinal Study to Assess the Quantitative Use of Fundus Autofluorescence for Monitoring Disease Progression in Choroideremia"

_jcm, 2021, doi:10.3390/jcm10020232_

Round 1

Reviewer 1 Report

General comments

A large clinical study including 80 choroideremia patients with a well-established causative gene mutation seems to be suitable for publishing in the Journal. The reviewer agree that PAF area proves to be useful in monitoring the rate of decline over long periods of time (>12 months minimum), but its use over the short term for assaying treatment responses may not be suitable. The observation period for some patients is relatively short (few months).

Specific comments

Line 69, “very versatile” is “versatile”?

Figure 5, it is difficult to distinguish between the three types of symbols.

Supple 1, same mutation seems to be siblings? It may be better to describe it in the foot note.

Reference, Hayakawa et al, Ophthalmic Genetics 1999 may be helpful concerning visual acuity and visual field, although it published more than 20 years ago.

Author Response

Thank you for your time reviewing this manuscript and provided the helpful feedback.

Line 69, “very versatile” is “versatile”?

Thank you to the reviewer for pointing out this over use of the word ‘very’, which we have now removed from line 69.

Figure 5, it is difficult to distinguish between the three types of symbols.

.  We have increased the contrast of the lightest points and increased the size of the symbols in the legend which we hope will improve the readability of this figure.

Supple 1, same mutation seems to be siblings? It may be better to describe it in the foot note.

Thank you for bringing up this point as it may be important for meta-analysis of this data. We have now put paired (sibling data) in the footnotes for the supplemental table. Each sibling group is in parenthesis with the subject numbers denoting the groups.

Reference, Hayakawa et al, Ophthalmic Genetics 1999 may be helpful concerning visual acuity and visual field, although it published more than 20 years ago.

Thank you for bringing this to our attention. This has now been added to the manuscript to support comments about acuity in the discussion. This addition has been added to line 326.

Reviewer 2 Report

The manuscript by Dubis et al. describes changes in visual acuity and PAF area using FAF imaging in a large cohort of patients with CHM. This study provides a longitudinal analysis of these changes and their correlation with the disease progression, taking into account the intra- and inter-observer variability. This type of analysis is of great interest of the scientific community as it provides new insights and key parameters for the use of FAF imaging in CHM prognosis and clinical trials. 

Minor consideration

-The authors need to clarify the type of tests they used to assess the statistical significance. 

Author Response

 We would like to thank the reviewer for their time looking through the manuscript and providing your feedback.

-The authors need to clarify the type of tests they used to assess the statistical significance. 

Thank you for identifying this point, we have now updated the manuscript to include this information. Specifically, we added to the statistics methods section the following information “Linear regressions were compared using nonparametric regression statistics and comparison between groups was done using the non-parametric Fischer’s Exact test.” This addition is at lines 149 and 150.
